# Sunflower Hybrids and Inbred Lines Adopt Different Physiological Strategies and Proteome Responses to Cope with Water Deficit

**DOI:** 10.3390/biom13071110

**Published:** 2023-07-12

**Authors:** Harold Duruflé, Thierry Balliau, Nicolas Blanchet, Adeline Chaubet, Alexandra Duhnen, Nicolas Pouilly, Mélisande Blein-Nicolas, Brigitte Mangin, Pierre Maury, Nicolas Bernard Langlade, Michel Zivy

**Affiliations:** 1INRAE UMR441, CNRS UMR2594, LIPME, Université de Toulouse, 31077 Toulouse, France; harold.durufle@inrae.fr (H.D.);; 2INRAE, ONF, BioForA, 45075 Orleans, France; 3AgroParisTech, GQE—Le Moulon, PAPPSO, Université Paris-Saclay, INRAE, CNRS, 91190 Gif-sur-Yvette, France; 4INRAE, INP-ENSAT Toulouse, UMR AGIR, Université de Toulouse, 31000 Toulouse, France

**Keywords:** drought, *Helianthus*, heterosis, proteomics, quantitative genetics, systems biology

## Abstract

Sunflower is a hybrid crop that is considered moderately drought-tolerant and adapted to new cropping systems required for the agro-ecological transition. Here, we studied the impact of hybridity status (hybrids vs. inbred lines) on the responses to drought at the molecular and eco-physiological level exploiting publicly available datasets. Eco-physiological traits and leaf proteomes were measured in eight inbred lines and their sixteen hybrids grown in the high-throughput phenotyping platform Phenotoul-Heliaphen. Hybrids and parental lines showed different growth strategies: hybrids grew faster in the absence of water constraint and arrested their growth more abruptly than inbred lines when subjected to water deficit. We identified 471 differentially accumulated proteins, of which 256 were regulated by drought. The amplitude of up- and downregulations was greater in hybrids than in inbred lines. Our results show that hybrids respond more strongly to water deficit at the molecular and eco-physiological levels. Because of presence/absence polymorphism, hybrids potentially contain more genes than their parental inbred lines. We propose that detrimental homozygous mutations and the lower number of genes in inbred lines lead to a constitutive defense mechanism that may explain the lower growth of inbred lines under well-watered conditions and their lower reactivity to water deficit.

## 1. Introduction

Heterosis (or hybrid vigor) is the phenomenon of a superior phenotypic performance of the hybrid progeny compared to its parental inbred lines. It has been described in many species and is exploited for the development of plant hybrid varieties. It can be observed not only for quantitative phenotypic traits controlled by multiple genes such as grain yield, plant biomass or height, but also for molecular traits such as protein or transcript relative abundance (e.g., [1,2,3]). Three non-exclusive genetic models have been proposed to explain heterosis: (i) dominance complementation, where hybrids benefit from the cumulative effect of all favorable dominant alleles inherited from both parents, (ii) overdominance, where the heterozygous state of a gene confers an advantage over the homozygous state, and (iii) epistasis, which implies a favorable interaction between the alleles brought together in the hybrid. Heterosis can be observed from an early developmental stage and can be influenced by the environment [4,5]. In support of the dominance complementation hypothesis, the combination in a hybrid of alleles that are adapted to different environments has been found to confer better homeostasis [6].

Sunflower (*Helianthus annuus* L.) is a major oilseed crop cultivated in drought-prone environments [7,8,9]. It ranks as the second most important hybrid crop after maize [10]. During the summer, sunflower is subjected to high evaporative demand and/or low soil water availability, especially in absence of irrigation. Different adaptive mechanisms are used by plants to maintain a favorable water status or tolerate dehydration [11]. An isohydric response occurs when stomatal control allows the maintenance of a high leaf water potential despite a declining soil water potential, whereas an anisohydric response occurs when stomatal conductance is reduced following a substantial decline in leaf water potential [12,13]. The responses of sunflowers to drought and their genetic variability have been documented at the levels of physiological traits, transcripts, proteins and metabolites [14,15,16,17,18,19,20,21]. Sunflower hybrids have been shown to respond differently from their parents to drought [22], but to our knowledge, the relationship between heterosis and drought response at these different levels of integration has never been examined. The objective of this study was to investigate the interplay between heterosis and drought responses in sunflower and analyze the relationship between protein expression and phenotypic traits. We used a systems biology approach combining the analysis of eco-physiological traits with that of proteomes from sixteen hybrids and eight parental lines grown under two different watering conditions. The parental genotypes included in this study produce different levels of heterosis and drought stress responses [23,24].

## 2. Materials and Methods

### 2.1. Plant Material and Growth Conditions

Full details of growth conditions and measurement of eco-physiological traits are given in the data paper describing this experiment [25]. In brief, experiments were carried out at the Heliaphen facility as described in Gosseau et al. [26]. Plants were grown in individual 15 L pots covered with a polystyrene sheet to prevent soil evaporation (Appendix A). Sixteen hybrids were grown alongside their eight parental inbred lines. The four female lines (SF009, SF092, SF109 and SF193, also known as ADV, IR, 2603RM and XRQ, respectively, according to the Sunflower Genetic Resource Center’s (INRAE) genotype nomenclature [27] are maintainer lines of PET1 cytoplasmic male sterility [28]; their sterile version was used to produce hybrids. The four male lines (SF279, SF317, SF326 and SF342, also known as OQP8, 83HR4RM, PSC8 and PW3RM, respectively) are male restorer lines. Each genotype was grown in triplicate under well-watered (WW) and water deficit (WD) conditions, resulting in a total of 144 plants. Irrigation of WD plants was stopped 38 days after germination (DAG). At that stage, plants had c. 20 leaves and were between growth stages R1 to R3 as defined by Schneiter and Miller [29], corresponding to the bud formation stage. Soil evaporation was estimated following Marchand et al. [30]. Plants were weighed three or four times a day by the Heliaphen robot to estimate transpiration [26]. Pairs of WD and WW plants were harvested when the fraction of transpirable soil water (FTSW) of WD plants reached 0.1 (here, between 42 and 47 DAG).

Three leaves, without their petioles, were harvested (between 11 a.m. and 1 p.m.) from each plant and immediately frozen in liquid nitrogen. Physiological traits (osmotic potential, carbon isotope discrimination, specific leaf area) were measured on two leaves representing two different developmental stages (referred to as LEAF_MATURE and LEAF_YOUNG). The first leaf (leaf number 16 ± 1.9) was positioned three-fifths (0.60 ± 0.04) of the height of the plant. It corresponded to a mature leaf near the end of its expansion and assumed to be experiencing its highest photosynthetic rate [31]. The second leaf (leaf number 20 ± 2.3) was positioned three-quarters (0.75 ± 0.07) of the height of the plant and four nodes above the mature leaf. This leaf was expanding and had a ~120 mm long blade (c. 50% of its final length). A third leaf, located just above the mature leaf, was sampled for the proteomic analysis.

The measured eco-physiological traits are listed in Table 1. In brief, FTSW was used as an indicator of the water deficit experienced by the plant [32]. Collar diameter and plant height were measured, and the number of leaves was counted at the beginning and end of the stress period. Rates were calculated by dividing the difference between values at two dates by the degree-day of this period. Leaf extension measurements represent expansion over the last 24 h. All other traits were measured at the end of the treatment.

### 2.2. Proteomics

Full methods for sample preparation, mass spectrometry, protein identification and quantification are described in Balliau et al. [33] and in the data paper describing the data used in the present paper [34]. In brief, leaf proteins were extracted using the TCA-acetone protocol [35]. Protein digestion and LC-MS/MS were performed as described in Balliau et al. [33]. Peptides were analyzed by LC-MS/MS using a Qexactive mass spectrometer (Thermo, Waltham, MA, USA). Protein identification was performed by searching the Heliagen database [36] using the X!Tandem search engine [37]. Data filtering and protein inference were performed using X!TandemPipeline 3.3.4 [38]. Only proteins with at least two different peptides with an e-value < 0.01 and a protein e-value < 10^−5^ were considered (other parameters are given in Appendix A). The false discovery rate at the peptide and protein level was 0.06% and 0%, respectively. Peptides were quantified based on extracted ion chromatograms (XIC) using MassChroQ [39], and intensity normalization was performed as described in Millán-Oropeza et al. [40]. Only proteins with at least two specific peptides that were quantified in at least 90% of the samples were retained for analysis. Relative protein abundance was calculated as the mean intensity of protein-specific peptides after subtraction of peptide-specific effects [41]. Functional annotations were obtained from the INRAE Sunflower Bioinformatics Resources (https://www.heliagene.org/HanXRQ-SUNRISE/downloads/1.2/20160309.HanXRQr1.0-20151230-EGN-r1.0.blast2go.zip (accessed on 10 July 2023)). Proteins were assigned to functional categories derived from Mapman controlled vocabulary. All LC-MS/MS data have been deposited on PROTICdb (http://moulon.inra.fr/protic/sunrise (accessed on 10 July 2023)).

### 2.3. Genotyping Procedure

Inbred lines were genotyped using the sunflower 50K SNP AXIOM array following the procedure described in Pecrix et al. [42] retaining heterozygote markers. In the end, 14,900 poly high resolution SNPs were considered for analysis. The heterozygosity rate in inbred lines ranged from 0.03% to 0.09%. Hybrid genotypes were deduced from the parental genotypes; hybrid heterozygosity rate ranged from 39.81% to 53.47%.

### 2.4. Statistical Analysis, Modelling and Data Integration

Data analysis was performed in the R environment (cran.r-project.org (accessed on 10 July 2023)). The effects of genotype (G, n = 24), treatment (T, n = 2) and genotype-treatment interaction (GxT) were tested by analysis of variance (ANOVA) for all eco-physiological traits and quantified proteins. For proteins, a Bonferroni correction for multiple comparisons was used (1211 proteins). Proteins were considered to have a significant effect when the corrected probability was <0.01 and the fold change between treatments (WD/WW) or between the highest and the lowest genotype was greater than 1.3 or lower than 1.3–1. This fold change threshold was used in all subsequent tests on proteins. All subsequent tests on protein levels were performed using the protein set selected here.

Heterosis tests were performed by testing deviation to additivity independently for each cross: contrasts between the mean value of two parental lines and the value observed in their hybrid were analyzed after analysis of variance. For proteins, multiple comparisons were corrected with a threshold of *p* < 0.0001. Heterosis by treatment interaction was assessed by analyzing the contrast between the deviation from additivity in WW and WD conditions.

Hybridity, i.e., the comparison of all hybrids and all parental lines, was tested with ANOVAs using hybridity status (line vs. hybrid) and treatment as factors. To take into account the correlation between replicates of the same genotype, the genotype was considered as a nested effect. As in the heterosis tests, a threshold of *p* < 0.0001 was used for proteins.

To analyze the response of up- and downregulated proteins, hybridity status x treatment (HxT) combinations were compared using a one-way ANOVA followed by a post-hoc Tukey test. HxT combinations were considered to be significantly different if the Tukey *p*-value was *p* < 0.05.

A cluster analysis was performed using the K-means method. Principal component analyses (PCA) and partial least square (PLS) regressions were performed using the mixOmics package v6.18.1 [43].

## 3. Results

### 3.1. Large Effect of Water Deficit on Physiological and Growth Traits

Twenty-one eco-physiological traits were measured in eight parental lines and 16 hybrids; 18 traits were measured at the end of the experiment and three were measured the day before the beginning of the treatment (details given in Table 1). Fourteen of the 18 traits measured at the end of the experiment showed a significant difference (*p* < 0.01) between WD (Water Deficit) and WW (Well-Watered) conditions (Appendix A). The four traits that showed no significant difference under water constraints were related to specific leaf area (SLA) and leaf area. While leaf area depended on leaf size before the treatment, leaf expansion rate measured expansion over the last 24 h and showed a significant reduction under WD in both mature and young leaves. A significant genotype effect was found for the majority of traits (*p* < 0.01 for 13 of the 21 traits). The number of plants analyzed for each genotype x treatment combination (n = 3) limited the detection of significant interaction; only one interaction was found to be significant (leaf area).

A principal component analysis (PCA) was performed to analyze the common variations of the different traits (Figure 1a). All plants subjected to WD were clearly separate from all those grown under WW conditions along the first principal component (PC1, 34% of the total variance). All traits contributed to PC1 (r > 0.5) (Figure 1b), except those measured prior to treatment plus leaf area and SLA. In addition, FTSW (fraction of transpirable soil water), which was not considered for the construction of the factorial axes but used as a stress indicator, was also strongly correlated with PC1 (r = 0.89) (Figure 1b). To better understand the response of the different genotypes to WD, we performed a second PCA on the differences between the values measured under the two conditions (WD minus WW) for all traits except those measured prior to treatment. In this analysis, PC1 (29% of the variance) separated inbred lines (negative values) from hybrids (positive values), showing the major importance of heterozygosity on the response to drought (Figure 1c). All traits associated with growth, with the exception of leaf area, were negatively correlated to PC1 (Figure 1d). This indicates that hybrids show a strong negative response compared with their parents. By contrast, SLA, CID (Carbon Isotope Discrimination) and the osmotic potential at full turgescence of young and mature leaves were positively correlated with PC1, indicating that the response of these traits was more positive and therefore reduced in hybrids compared to inbred lines. These results can be interpreted as a stronger physiological response (more osmotic adjustment, more carbon discrimination and increased leaf thickening) and a reduced impact of drought on growth in lines compared to their hybrid offspring.

### 3.2. Hybrids Show Heterosis for Eco-Physiological Traits Measured before Treatment and Differ from Inbred Lines in Their Response to Water Deficit

The second component of the PCA (PC2, 21% of the total variance, Figure 1a) separated hybrids from parental lines and was mostly determined by traits measured prior to treatment and leaf area: compared to their parents, hybrids were taller and had a larger collar diameter and a larger leaf area with more leaves before treatment, although individually their leaves were smaller (Figure 1b). This result suggests the presence of a stronger heterosis effect on these traits than on the traits measured after treatment. This hypothesis was confirmed by statistically testing for a heterosis effect in each of the 16 hybrids (Appendix A). Indeed, heterosis was significant (*p* < 0.01) in 9 to 12 hybrids for the three traits measured before treatment (collar diameter, plant height and number of leaves) and in 0 to 5 hybrids (1.9 on average) for the other traits.

Four hybrids exhibited a significant heterosis × treatment interaction (HxT) for mature leaf expansion and plant height growth rate, and three hybrids exhibited a significant HxT for leaf expansion rate (Appendix A). For the remaining traits, only 0 to 1 hybrid (0.9 on average) exhibited a significant heterosis × treatment interaction. Thus, only limited effects of heterosis × treatment interactions were detected. Nevertheless, the PCA of differences measured between WD and WW conditions (Figure 1c) showed a clear separation between lines and hybrids, suggesting there is heterosis in response to WD. To confirm that hybrids and lines formed two populations that differed in their response to WD, we tested each trait for the effect of hybridity (hybrids vs. inbred lines) and hybridity × treatment interaction (Appendix A). Ten traits showed a significant hybridity effect (*p* < 0.01) including the three traits measured before treatment, and six traits showed a significant hybridity × treatment interaction. Eleven of the traits measured after treatment showed either a hybridity or hybridity × treatment interaction effect.

To determine if the traits or trait responses to WD measured in hybrids were related to the genetic distance between parents, we performed regression analyses using the rate of heterozygosity in hybrids as a proxy for the genetic distance between parents. No trait was affected by the rate of heterozygosity in hybrids or by its interaction with the treatment (*p* < 0.01) (Appendix A). This indicates that the genetic distance between parents had little or no effect on the traits or trait responses to WD observed in hybrids.

Taken together, these results show that, while only a few eco-physiological traits showed significant mid-parent heterosis, hybrids and lines constituted two distinct populations differing both in their phenotype in the absence of WD and in their response to WD. This is clearly visible in Figure 2, which shows that the better performance of hybrids before treatment for plant height, collar diameter and leaf number decreased under WD. Furthermore, growth traits, osmotic potential and transpiration decreased more in hybrids than in parental lines under WD, while the opposite was observed for carbon isotope discrimination and osmotic potential at turgescence. This indicates that osmotic adjustments and stomatal responses were higher in parental lines than in hybrids.

### 3.3. Distinct Populations of Proteins Are Up- and Downregulated in Sunflower Leaves in Response to Water Deficit

LC-MS/MS analysis of proteins in mature leaves resulted in the identification of 3062 different proteins (Appendix A), of which 1211 were quantified. Graphical representation of their variation according to genotype and treatment for all crosses is provided in Appendix A. Of these, 471 differentially abundant proteins (DAPs) showed either a significant treatment (n = 256; 108 upregulated and 148 downregulated proteins), genotype (n = 285) or genotype x treatment effect (n = 2) (Appendix A, column X). These DAPs were distributed into 12 clusters using the K-means method and comprising 22 to 54 DAPs (Appendix A). The DAPs in clusters 1, 8 and 12 were clearly downregulated and those in clusters 3, 6 and 11 were upregulated (Appendix A). The DAPs in cluster 6 also showed a large genetic effect. The DAPs in clusters 5, 7 and 10 showed a significant treatment effect, but with an R^2^ between 2 and 6%. Functional categories were not evenly distributed among these clusters. For example, major carbohydrate (CHO) metabolism was mostly represented by cluster 8 (9 out of 14 proteins), protein folding by cluster 11 (11 out of 31 proteins), transcription by cluster 1 (8 out of 18 proteins), tetrapyrrole synthesis by cluster 1 (7 out of 8 proteins), and translation by cluster 12 (11 out of 25 proteins, Appendix A).

Among upregulated proteins, the most represented functional categories were redox homeostasis and protein folding (Figure 3). The ten most upregulated proteins were involved in redox homeostasis, detoxification, amino acid degradation and minor CHO metabolism (Appendix A). Among these, the protein with the highest fold change (11.4) is of unknown function and annotated as “nodulin-related”. The other most upregulated proteins (fold change > 3) were heat shock proteins (HSPs), late embryogenesis abundant proteins (LEAs), a non-specific lipid transfer protein (LTP), a sucrose synthase, an auxin-repressed dormancy-associated protein and a kirola-like MLP43 (major latex protein). Several of the other upregulated proteins were directly related to known responses to drought stress such as stomatal closure (aspartic protease in guard cells 1) [44], GABA (glutamate decarboxylase) synthesis [45] and osmoprotection, including mitochondrial delta-1-pyrroline-5-carboxylate dehydrogenase [46,47], sorbitol dehydrogenase [48,49], betaine aldehyde dehydrogenase and mannitol dehydrogenase.

Downregulated proteins were mostly involved in transcription and translation, amino acid synthesis and CHO (mostly starch) metabolism. Remarkably, they included seven of the eight quantified proteins involved in tetrapyrrole or chlorophyll synthesis and six of the eight proteins involved in cell organization. Interestingly, two ATP sulfurylases and a sulfite reductase, which are involved in the first steps of sulfur assimilation, were downregulated.

### 3.4. The Proteome Shows Constitutive and Water Status-Dependent Heterosis

PCA of the abundance of the 471 DAPs showed a clear separation between WW and WD conditions along PC1 (32% of the total variance) (Figure 4a), which was greater for hybrids than for parental lines (with the exception of SF009). This result indicates that, at the proteome level, hybrids display a more pronounced response to WD than inbred lines. These trends are also highlighted in Figure 4c, where genotypes are separated according to their response to WD. The DAPs in clusters 1, 3, 6, 8, 11 and 12 mostly contributed to this differentiation (Figure 4b,d), indicating that they were involved in the hybridity × treatment interaction. This is supported by the results presented in Figure 5, showing that for clusters 1, 3, 6, 8, 11 and 12, hybrids showed more reactivity to WD than parental lines.

PCA also showed that lines and hybrids were partially separated along PC2 (13% of the total variance) (Figure 4a), which was positively correlated with the DAPs in clusters 2 and 4 (Figure 4b). These two clusters comprised proteins that were more abundant in lines than in hybrids (Appendix A). Taken together, they comprised 10 of the 27 proteins involved in central carbon metabolism, four proteins involved in stress responses, seven proteins involved in photosynthesis and five proteins involved in redox homoeostasis. The DAPs in clusters 7 and 9 were negatively correlated with PC2 (Figure 4b). No particular enrichment was observed in these two clusters. Since no other principal component showed a separation between hybrids and lines (not shown), the results shown in Figure 4a suggest that hybridity may represent less variation than the hybridity × treatment interaction.

To confirm these results, heterosis and heterosis × treatment interaction were tested for each protein independently in the 16 hybrids, as well as the effect of hybridity and hybridity × treatment on the whole set of genotypes. Significant effects of mid-parent heterosis were found in 63 cases (i.e., protein × hybrid combinations). They involved 43 proteins for which no functional enrichment was observed. Among these, nine were significantly upregulated and five were significantly downregulated in response to WD. For each protein, the number of hybrids showing mid-parent heterosis varied from 1 (for 27 proteins) to 4 (for 1 protein). The progeny of SF326 displayed the most heterosis for protein abundance (Appendix A) but this was not associated with genetic distance as SNP (Single Nucleotide Polymorphism) data showed that this parental line was not the most genetically distant line (Appendix A). The number of cases of positive and negative heterosis was similar (respectively 33 and 30), with an average fold change (the hybrid value divided by the mean parental value) of 2.7 and 0.52, respectively. Significant heterosis × treatment interaction was found in 30 cases involving 29 proteins. Although this interaction was significant in around one hybrid per protein, the trend was often visible in several other hybrids (see Appendix A). Among these 29 proteins, 18 were upregulated and 9 were downregulated in response to WD.

Finally, 2 and 27 proteins showed significant hybridity and hybridity × treatment effects, respectively. Of the latter, 13 were among those showing a significant heterosis × treatment effect in at least one hybrid.

To study more precisely the differences in response to WD between hybrids and lines, we performed a one-way ANOVA with the combination of hybridity status and treatment as the unique factor, followed by a *post hoc* Tukey test (Appendix A). SF009 was excluded from this analysis. Forty-five of the upregulated proteins were more abundant in hybrids than in lines under WD conditions (i.e., the value was significantly higher in hybrids than in lines under WD condition according to the *post hoc* Tukey test), whereas only two proteins were more abundant in the parental lines. Similarly, 29 of the downregulated proteins were less abundant in hybrids than in lines under WD conditions, whereas no protein was less abundant in the parental lines. Taken together, these statistical tests confirm the trends observed from principal component and cluster analyses: at the proteome level, hybrids respond more strongly than lines to WD. Remarkably, three of the most upregulated proteins (the nodulin-related protein, the auxin-repressed protein and the embryonic DC-8 protein) were more upregulated in hybrids than in lines.

Differences in the reactivity of hybrid and line proteomes are partly due to pre-existing differences that can be seen under WW conditions.

Differences in reactivity between hybrids and lines may be related to the protein levels reached under WD conditions but may also be related to differences existing under WW conditions. Using the same analysis as above (a one-way ANOVA and a Tukey test), we tested whether significant differences existed between hybrids and lines under WW conditions for up- and downregulated proteins. Under WW conditions, nine of the upregulated proteins were significantly more abundant in lines than in hybrids, while none were more abundant in the hybrids (Appendix A). Similarly, 35 of the downregulated proteins were less abundant in lines than in hybrids, while the opposite was true for only one protein. These results show that under WW conditions, 44 proteins were closer to their level under WD conditions in lines than in hybrids, while the opposite was true for only one protein. The other up- or downregulated proteins showed no significant difference between hybrids and lines under WW conditions. Overall, under WW conditions, up- and downregulated proteins were either closer to their level under WD conditions in lines than in hybrids, or not significantly different in lines and hybrids. Practically no protein (only one) was closer to its level under WD conditions in hybrids than in lines. This may be because lines are more stressed under WW conditions. Among the nine upregulated proteins with higher levels in lines than in hybrids under WW conditions, two have functions associated with protection against stress: a peptide methionine sulfoxide reductase (HanXRQChr08g0223071) and a glutamate decarboxylase (HanXRQChr07g0190181). However, not all functional categories generally associated with stress seem to contribute to this stressed state: for example, most upregulated proteins involved in protein folding (11 out of 14) are in cluster 11, where proteins are not at a higher level in lines than in hybrids under WW conditions (Appendix A). Similarly for downregulated proteins, cluster 8 (containing proteins that show no difference between lines and hybrids under WW conditions) includes nine proteins involved in major CHO metabolism, while clusters 1 and 12 contain only one protein belonging to this category. By contrast, most proteins involved in protein translation and transcription are in clusters 1 and 12, i.e., they are less abundant in lines than in hybrids under WW conditions.

### 3.5. Co-Variation between Eco-Physiological Traits and Protein Abundances

The relationship between DAPs and eco-physiological traits under WD and WW conditions was analyzed using partial least square regressions (PLS) (Figure 6a,b). PC1 separated the WW and WD conditions, and logically associated physiological traits that were affected by WD to DAPs affected by WD and belonging to clusters 1, 3, 6, 8, 11 and 12. PC2 separated hybrids and lines and was essentially determined by traits that showed no or little effect of WD and by DAPs belonging to cluster 2 (median correlation: 0.52) for which no significant enrichment was detected. To study this relationship between eco-physiological traits and DAPs more finely beyond this obvious observation, a PLS analysis was performed on trait and protein responses (Figure 6c,d). Here, PC1 separated lines from hybrids, while PC2 separated line SF009 from all the other genotypes. All traits related to growth, transpiration and osmotic potential as well as DAPs belonging to clusters 1, 8 and 12 (downregulated DAPs) were negatively correlated with PC1. Under WD, these traits and DAPs decreased more strongly in hybrids than in lines. The traits that were positively correlated with PC1 either decreased more strongly in lines than in hybrids (i.e., osmotic potential at turgescence and CID) or were not significantly affected by WD. The DAPs that were the most positively correlated with PC1 were upregulated proteins belonging to clusters 3 and 11, containing proteins that are more upregulated in hybrids than in lines.

### 3.6. The Response of Physiological Traits Is Partly Related to the Proteome under WW Conditions

The fact that the differences in response to WD are partly related to protein levels under WW conditions makes it possible to predict, in part, the proteome response to WD from the proteome of plants grown under WW conditions. Indeed, the correlation between clusters 1, 3 and 12 in WW plants and their response to WD are, respectively, −0.92, −0.71 and −0.72. In several instances, the values of a cluster in the WW condition are correlated with the response of other clusters: for example, the correlation between the values of cluster 3 (upregulated DAPs) in WW and the response of cluster 12 (downregulated DAPs) is 0.66 (Appendix A). This observation led us to investigate a possible link between protein amounts under WW conditions and the response of physiological traits. We focused on upregulated proteins, the hypothesis being that these proteins could contribute to an anticipated protection against a possible future stress (Figure 7). PLS analysis performed on the proteomics data collected under WW conditions and the eco-physiological responses to WD showed that some DAPs belonging almost exclusively to clusters 3 and 6 were highly negatively correlated with PC1 (correlation < −0.5) (Figure 7b). Among the DAPs that were the most negatively correlated with PC1, we found an endo-1,3 1,4-beta-D-glucanase-like protein, which is involved in cell wall growth [50], a phosphoethanolamine N-methyltransferase, which is a key enzyme for glycine-betaine synthesis, and a carotenoid cleavage dioxygenase highly similar to NCED, a key enzyme in the synthesis of abscisic acid (Appendix A). As the responses of growth traits were also negatively correlated with this component, we can consider that the abundance of these proteins in WW plants is associated with a low effect of WD on growth reduction, i.e., a higher growth maintenance, which was mainly observed in lines. Most of the other upregulated proteins (mainly from cluster 11) showed a weak positive correlation with PC1, and are thus not related to or are slightly correlated with the growth reduction response to WD. As previously mentioned, compared to clusters 3 and 6, cluster 11 is enriched in proteins involved in protein folding (25% vs. 4%). Overall, most of the proteins belonging to categories that are generally associated with stress (protein folding, detoxication, redox homeostasis, stress) were not among the proteins that were the most negatively correlated with component 1 (Appendix A). Thus, their constitutive abundance in WW conditions cannot be associated with growth maintenance under WD.

## 4. Discussion

This study assessed the impact of heterozygosity on the response of sunflower to water deficit and the molecular processes involved by comparing the phenotypes and proteomes of eight sunflower parental lines and their 16 hybrids grown under WW and WD conditions.

Under the WW condition, all growth traits were higher in the hybrids than in the parental lines. Higher growth in hybrids was accompanied by a higher transpiration rate and a slightly lower osmotic potential than in the parental lines. Hybrid leaves were slightly smaller than those of parental lines, but as their number was greater, total leaf area was on average higher in hybrids. Thus, under the WW condition, hybrids displayed a “profligate/opportunistic” strategy, i.e., productive profligate water use patterns that allow them to maximize photosynthesis and growth when water is available [51,52,53]. Compared to hybrids, parental lines showed a “conservative” strategy associated with water conservation and reduced growth that ensures survival in case of drought at the expense of productivity [11].

Under WD, both hybrids and parental lines decreased their vegetative growth, reduced their transpiration, performed osmotic adjustments and improved their water use efficiency (decreased carbon isotope discrimination). For all traits related to growth as well as for transpiration, hybrids responded to WD more sharply than parental lines. By strongly reducing their growth, hybrids tended to limit the decrease in leaf water potential, but this response was not entirely successful since leaf osmotic potential decreased more in hybrids than in parental lines. Osmotic potential at turgescence decreased more in lines than in hybrid. Accordingly, leaves in hybrids achieved a lower level of osmotic adjustment than those of parental lines, while being submitted to a higher osmotic stress. Overall, hybrids were characterized by a strong growth response and a somewhat lower physiological response compared to parental lines (osmotic adjustment, stomatal closure and to a lesser extent leaf thickening). It must be pointed out, however, that in a natural environment, hybrids can extract water from deeper soil layers (>40 cm depth) during a dry period [53], which can contribute to drought resistance. Even so, we found highly different water management strategies between sunflower hybrids and their parental lines. This is consistent with the genotypic differences previously reported in other species. For example, anisohydric and isohydric behaviors have been described in different grapevine cultivars [54] and in different poplar genotypes [55]. However, our study is the first to establish in a crop a relationship between water management strategy and hybridity status. It is worth noting that the genetic distance between the two parental lines did not explain the response level to drought of the corresponding hybrid.

At the molecular level, water stress strongly impacted the leaf proteome: about 20% of quantified proteins were significantly up- or downregulated under WD. Many of the upregulated proteins were involved in functions known to be activated in response to water deficit (chaperones, redox homeostasis, detoxification, osmoprotectant synthesis), while downregulated proteins were mostly involved in protein synthesis, carbohydrate synthesis and photosynthesis. All genotypes responded to water deficit by triggering defense mechanisms against stress and by reducing metabolic functions associated with growth.

Analysis of the 16 crosses showed that 6% of quantified proteins displayed significant mid-parent heterosis or heterosis x treatment interaction. For most of these proteins, heterosis was significant for a small number of hybrids (37% on average). In the leaves of one maize hybrid, Birdseye et al. [56] found that 32% of proteins exhibited heterosis. However, using the minimum fold change considered here (±30%) this percentage falls to 6%, which is similar to what has been found in other studies of plants and yeast (2 to 13%) [1,2,57,58,59]. Positive and negative heterosis were equally represented (52% and 48%, respectively), which is also consistent with what has been previously observed in intra-specific crosses in plants and yeast [1,2,57,59]. Therefore, our study confirms the trend observed in other organisms, i.e., heterosis for protein abundance involves a relatively low number of proteins and is cross-dependent, and positive and negative heterosis are found in similar measure in intraspecific crosses.

Although our statistical analyses lacked the power to detect heterosis x treatment interactions in individual crosses, comparison of the selected lines and their hybrids revealed that these exhibited very different responses to drought. Indeed, hybrids clearly showed a stronger response to drought than lines, with respect to downregulated as well as upregulated proteins. Overall, upregulated proteins were more upregulated in hybrids than in lines and downregulated proteins were more downregulated in hybrids than in lines. This difference in reactivity was partly due to differences in protein abundance between hybrids and lines in the WW condition: some upregulated (downregulated) proteins were less (more) abundant in hybrids than in lines in the WW condition. By contrast, almost no upregulated (downregulated) protein was more (less) abundant in hybrids than in lines in the WW condition.

The differences observed between hybrids and lines at the proteome level may correspond to different strategies in response to water deficit. By devoting fewer resources to protein synthesis and photosynthesis and more resources to stress responses under WW conditions, parental lines may maintain themselves in a preventive state that may allow them to respond better to a possible stress than hybrids. By contrast, by devoting more resources to the metabolism necessary for rapid growth, hybrids appear more profligate than parental lines in WW conditions. As a consequence, hybrids may need to be more responsive to withstand WD. This hypothesis is in agreement with the two response strategies observed at the phenotypic level.

Upregulated proteins were negatively correlated with plant growth, as they were most abundant in plants that reduced their growth the most (hybrids under WD conditions). This negative correlation was also stronger for proteins that were more abundant in lines (slow growth) than in hybrids (fast growth) under WW conditions. Many of these proteins protect against cell damage caused by water stress (by maintaining redox homeostasis, protecting native protein conformations, osmoprotection, etc.). While they do not contribute to growth (or possibly contribute to reduce growth), they probably favor the recovery of growth processes after WD by maintaining cells in a functional state. This may be the case for upregulated proteins involved in protein folding, which showed no difference between hybrids and lines under WW conditions, or proteins involved in the reduction of oxidized protein methionines and GABA metabolism, which were significantly more abundant in lines than in hybrids under WW conditions. Further investigation of the recovery capability of hybrids and lines could be performed to test these hypotheses.

Nevertheless, despite being negatively correlated with growth, the higher constitutive expression of some upregulated proteins could contribute to maintain growth under WD. PLS analysis showed that there was a positive correlation between the abundance of some upregulated proteins under the WW condition and the maintenance of growth under WD. Two of the best correlated proteins are involved in cell growth and the synthesis of the osmoprotectant glycine-betaine.

Overall, our results suggest that there is a trade-off between growth and defense mechanisms. Such a trade-off has been observed previously for transcripts [60,61] and metabolites [62]. Groszmann et al. [63] also found that defense genes have a lower basal expression in Arabidopsis hybrids than in inbred lines. Moreover, Miller et al. [64] found that in Arabidopsis grown in non-stressful conditions, stress-responsive genes were repressed in hybrids compared to their parents. They also found a significant correlation between the expression of stress-responsive genes in the parental lines and heterosis for biomass. They suggested that the constitutive expression of the different stress-defensive genes inherited from the parental lines would be detrimental to hybrid growth. In their study, abiotic stress triggered the expression of stress genes, which were sometimes expressed at a higher level and with a different timing in Arabidopsis hybrids compared to their parental lines. A time course study would be necessary to establish whether the higher abundance of stress-related proteins in sunflower hybrids under WD is combined with a shift in the timing and faster regulation of gene expression. In maize, Birdseye et al. [56] observed a relationship between heterosis for the amount of chloroplastic ribosomal proteins and heterosis for plant height. The results presented here are different, maybe because of differences between C4 plants (such as maize) and C3 plants (such as sunflower): heterosis was not found here for either cytoplasmic or chloroplastic ribosomal proteins. Almost all proteins were more downregulated in hybrids than in lines and their variation was not correlated with growth reduction. However, further shared molecular mechanisms between plant species might be observed when studying non-photosynthetic organs involved in drought response in key organs such as roots and stems.

With the exception of the interaction with treatment described above, no significant relationship between heterosis for protein abundance and eco-physiological traits was found. A relationship between gene or protein expression and phenotypic traits is often difficult to establish because of the non-linear relationship between traits at different levels of integration. Note that this non-linearity can also be a cause of heterosis [65,66]. Different general mechanisms have been proposed to explain heterosis in gene expression [67,68], but they do not explain the specific regulation of proteins involved in the stress response we observed here.

## 5. Conclusions

In conclusion, our results show that under WD conditions, sunflower hybrids reduce their growth more rapidly than inbred lines, and that their leaf proteome is more responsive to WD. These results indicate the existence of a trade-off between defense and growth mechanisms that would be managed differently by hybrids and inbred lines: in the absence of stress, hybrids would favor growth to the detriment of defense, while inbred lines would reduce their growth to maintain constitutive defense mechanisms; in the presence of stress, hybrids would favor defense to the detriment of growth, while inbred lines would maintain a balance between growth and defense.

We explain most of our results by the constitutive stress suffered by inbred lines. Sunflower is an allogamous species which shows considerable inter-individual gene number variations. Sunflower inbred lines can contain thousands of specific presence/absence variants [69], and thus hybrids potentially contain thousands more genes than their parents. Gene loss and homozygosity of detrimental alleles would induce stress in inbred lines, which would respond by increasing the synthesis of stress proteins and decreasing the synthesis of proteins implicated in normal growth (protein synthesis, photosynthesis). These molecular responses would finally translate into a growth reduction at the plant level. The lower response to WD of inbred lines compared to hybrids could be due to their inability to respond adequately to drought. It could also be due to the protection conferred by higher levels of stress proteins accumulated under WW conditions, which may play a preventive role.

## Figures and Tables

**Figure 1 biomolecules-13-01110-f001:**
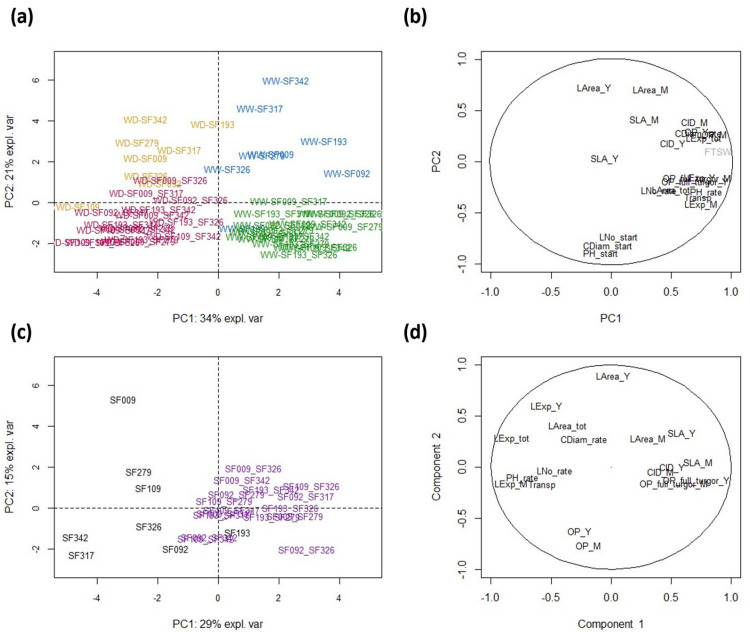
Comparison of eco-physiological traits in 8 sunflower parental lines and their 16 hybrids under well-watered (WW) and water deficit (WD) conditions. (**a**) Principal component analysis (PCA) of the eco-physiological traits (Lines-WW in blue, Lines-WD in orange, Hybrids-WW in green and Hybrids-WD in red) and (**b**) the correlation circle plot. (**c**) PCA of the responses of eco-physiological traits to WD (WD-WW; Lines in black and Hybrids in purple) and (**d**) the associated correlation circle plot. Hybrid names are the concatenation of the female and male names of their parental lines.

**Figure 2 biomolecules-13-01110-f002:**
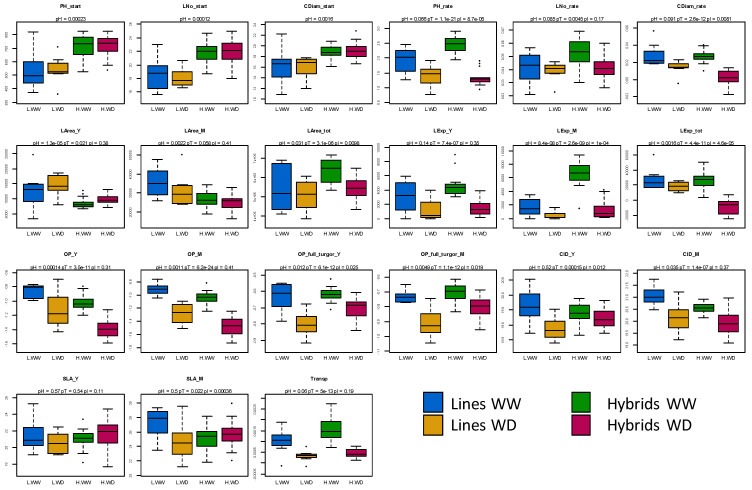
Variation of 21 eco-physiological traits in 8 sunflower parental lines and their 16 hybrids according to water and hybridity status. Lines-WW in blue, Lines-WD in orange, Hybrids-WW in green and Hybrids-WD in red. pH: *p*-value of the hybridity effect, pT: *p*-value of the treatment effect and pI: *p*-value of hybridity x treatment interaction (HxT).

**Figure 3 biomolecules-13-01110-f003:**
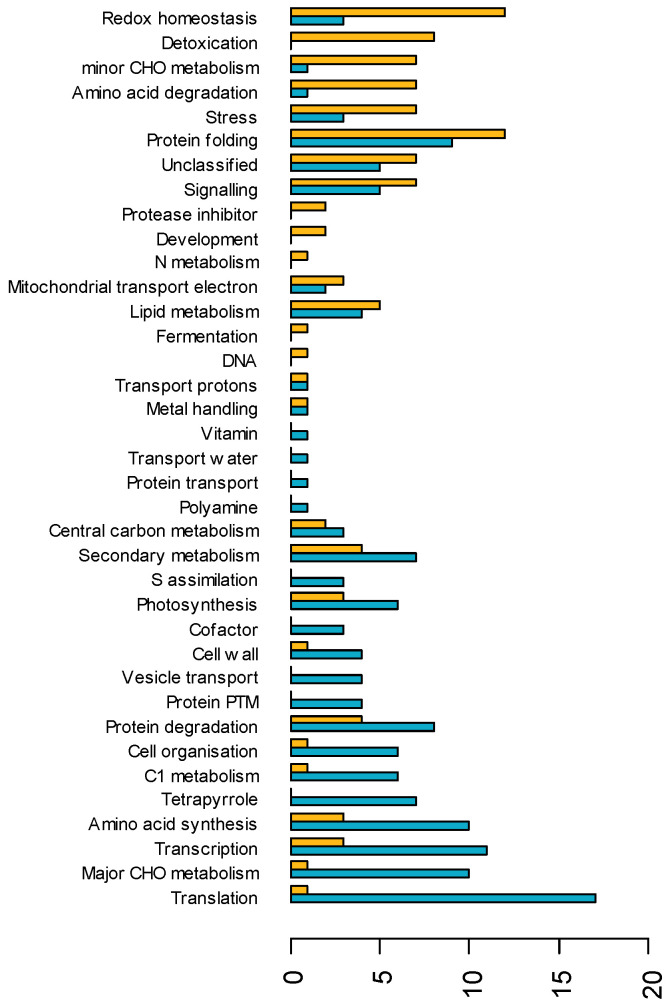
Functional category of downregulated (blue) and upregulated (orange) proteins in response to WD.

**Figure 4 biomolecules-13-01110-f004:**
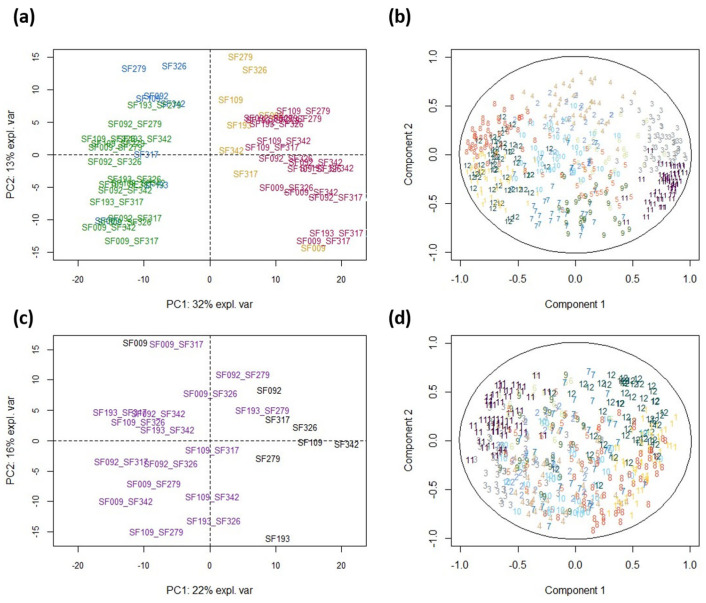
Comparison of the leaf proteomes of 8 sunflower parental lines and their 16 hybrids under WW and WD conditions. (**a**) PCA of 471 differential abundant proteins (DAPs) (Lines-WW in blue, Lines-WD in orange, Hybrids-WW in green and Hybrids-WD in red), and (**b**) the associated correlation circle showing the correlation of DAPs with components 1 (PC1) and 2 (PC2) of the PCA, (**c**) PCA of the response of DAPs to WD (WD-WW; Lines in black and Hybrids in grey) and (**d**) the associated correlation circle plot. DAPs are color-coded according to their cluster membership.

**Figure 5 biomolecules-13-01110-f005:**
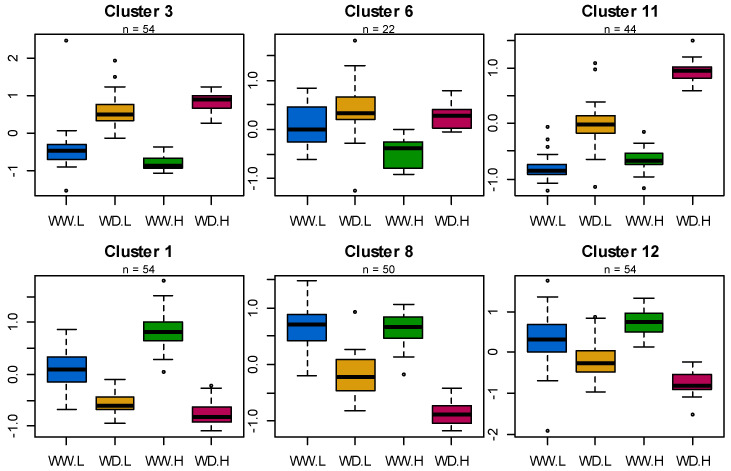
Response of the 6 protein clusters showing a significant treatment effect to hybridity and water status. Hybridity status: hybrid vs. line (H/L); water status: WD/WW. Lines-WW in blue, Lines-WD in orange, Hybrids-WW in green and Hybrids-WD in red (the SF009 line was excluded, see text).

**Figure 6 biomolecules-13-01110-f006:**
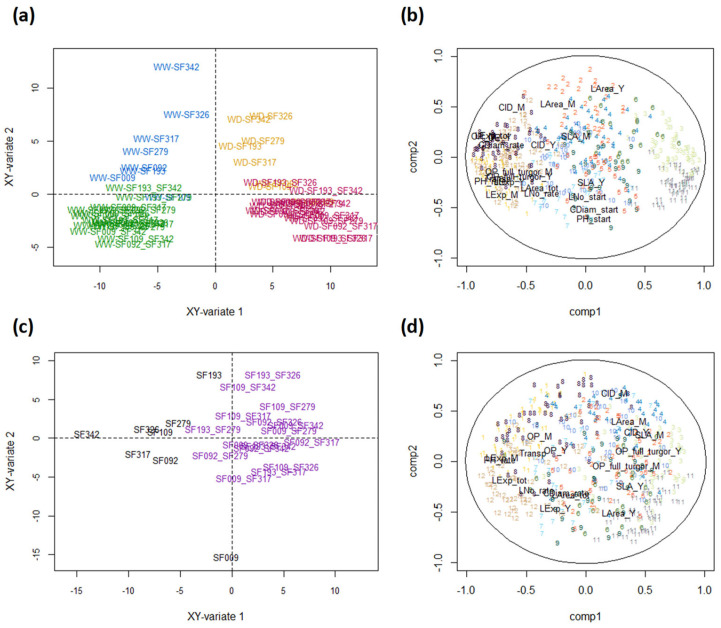
(**a**) Separation according to the Partial Least Squares regression (PLS) of DAPs and eco-physiological traits (Lines-WW in blue, Lines-WD in orange, Hybrids-WW in green and Hybrids-WD in red), and the (**b**) associated correlation circle. (**c**) Hybridity status separation according to the PLS analysis of DAP and eco-physiological trait responses to WD (WD-WW; Lines in black and Hybrids in purple), and the (**d**) associated correlation circle. In the correlation circles, DAPS are color-coded according to their cluster membership and eco-physiological variables are in black.

**Figure 7 biomolecules-13-01110-f007:**
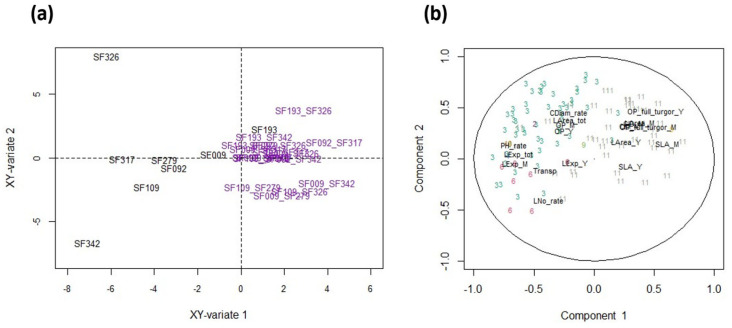
Integrative analysis of drought responses of DAPs and eco-physiological traits: PLS analysis of the upregulated DAPs under WW conditions and the response of the eco-physiological traits. (**a**) Separation between genotypes; Lines in black and Hybrids in purple and (**b**) the associated correlation circle plot. DAPs are color-coded according totheir cluster membership and the eco-physiological variables are in black.

**Table 1 biomolecules-13-01110-t001:** List of environmental and ecophysiological variables.

Category	Level of Organisation	Trait	Abbreviation	Unit	Measurement Time
Water deficit experienced by the plant	Plant	Fraction of transpirable soil water	FTSW	(N/A)	End of exp.
Morphology	Plant	Plant height	PH_start	(mm. plant^−1^)	37 DAG
		Number of leaves	LNo_start	(plant^−1^)	37 DAG
		Collar diameter	CDiam_start	(mm. plant^−1^)	37 DAG
		Leaf area	LArea_tot	(mm^2^. plant^−1^)	End of exp.
	Leaf	Mature−leaf area	LArea_M	(mm^2^)	End of exp.
		Young−leaf area	LArea_Y	(mm^2^)	End of exp.
		Mature Specific leaf area	SLA_M	(m^2^. kg^−1^)	End of exp.
		Young Specific leaf area	SLA_Y	(m^2^. kg^−1^)	End of exp.
Growth	Plant	Plant height growth rate	PH_rate	(mm. °Cd^−1^)	From 35 DAG to the end of exp.
		Rate of leaf appearance	LNo_rate	(leaves. °Cd^−1^)	From 35 DAG to the end of exp.
		Collar diameter growth rate	CDiam_rate	(mm. °Cd^−1^)	From 35 DAG to the end of exp.
		Leaf expansion rate	LExp_tot	(mm^2^. d^−1^)	Last 24 h of the exp
	Leaf	Mature-leaf expansion rate	LExp_M	(mm^2^. d^−1^)	Last 24 h of the exp.
		Young-leaf expansion rate	LExp_Y	(mm^2^. d^−1^)	Last 24 h of the exp.
Physiology	Plant	Plant transpiration rate	Transp	(g. mm^−^^2^. d^−1^)	Last 24 h of the exp.
	Leaf	Osmotic potential of mature leaf	OP_M	(MPa)	End of exp.
		Osmotic potential of young leaf	OP_Y	(MPa)	End of exp.
		Osmotic potential at full turgor of mature leaf	OP_full_turgor_M	(MPa)	End of exp.
		Osmotic potential at full turgor of young leaf	OP_full_turgor_Y	(MPa)	End of exp.
		Carbon isotope discrimination of mature leaf	CID_M	(‰)	End of exp.
		Carbon isotope discrimination of young leaf	CID_Y	(‰)	End of exp.

## Data Availability

The raw data, intermediate files, extraction methods, and all biological material examined in this study are available and fully described in the data papers Blanchet et al. [25] and Balliau et al. [34].

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
