# Peer review of "Sunflower Hybrids and Inbred Lines Adopt Different Physiological Strategies and Proteome Responses to Cope with Water Deficit"

_biomolecules, 2023, doi:10.3390/biom13071110_

Round 1

Reviewer 1 Report (Previous Reviewer 1)

The manuscript has been significantly improved and now warrants publication in Biomolecules.

Author Response

The manuscript has been significantly improved and now warrants publication in Biomolecules.

Thank you

Reviewer 2 Report (New Reviewer)

The manuscript investigates the differences in the physiological response to drought stress of pure sunflower lines versus their hybrids, through a system biology approach

The manuscript is well written and data are clearly reported. The work plan is clear and analysis have been conducted in a correct way, followed by solid statistical analysis. Some minor mistakes here and there are highlighted in the revised text and the mains are reported below. Some inconsistency between the main text and Figures was also found.

The first issue is related to the title, I believe that the term proteomic is misused here, it doesn’t sound correct to me to speak about “proteomic strategies” of plants, rather to “proteome (or proteomic) responses”. The proteome variation is just a consequence of the physiological strategy. The term ‘proteomic strategies’ seems to me more suited to indicate different ways to investigate the proteome. I would suggest “physiological strategies and proteome response”

Line 24: What do the Authors mean by “absence of genes” in inbred lines? Which genes? Please clarify.

Line 24: presence/absence of polymorphism

Line 73: Three traits are reported to be measured the day before the beginning of the treatment, 35 DAG in Table S1. However, in the Method’s section, the beginning of the treatment is reported at 38 DAG (line 475). Please explain or fix this inconsistency.

Line 160: the category transcription is reported in Table S8 (and S6 and Figure 3) as RNA, please uniform.

Line 162: I would say 2,3,7 and 11, from Table S8

Line 217:

Table S1: I would suggest moving this Table to the main text, as it also reports all the abbreviations used in Fig.2. The number of eco-physiological traits here reported is 22 (column D), while 21 are reported in the title and in the main text (line 71). This likely because FTSW was just used as the measure of the watering state, but the information must be consistent. Heading of column E, ‘abbreviations’ is missing a ‘b’. I think the correct English abbreviation for number is No. or No, not Nb as in French (LNb_start). Same in Table S3, column H and I and elsewhere.

Table S3: I think it would be more easy to follow the text if the three traits measured before the start of the treatment are reported separated from the other traits, i.e. in the first three row, or highlighted in some way (boldface or italics).

Table S8: the total number of proteins in each Functional category could be indicated in a last column to simplify reading of the corresponding test. The last table line has no heading

Names of Tables S8 and S9 are exchanged in the excel file

Author Response

The manuscript investigates the differences in the physiological response to drought stress of pure sunflower lines versus their hybrids, through a system biology approach

The manuscript is well written and data are clearly reported. The work plan is clear and analysis have been conducted in a correct way, followed by solid statistical analysis. Some minor mistakes here and there are highlighted in the revised text and the mains are reported below. Some inconsistency between the main text and Figures was also found.

=>Thank you very much for your thorough reading of the manuscript.

The first issue is related to the title, I believe that the term proteomic is misused here, it doesn’t sound correct to me to speak about “proteomic strategies” of plants, rather to “proteome (or proteomic) responses”. The proteome variation is just a consequence of the physiological strategy. The term ‘proteomic strategies’ seems to me more suited to indicate different ways to investigate the proteome. I would suggest “physiological strategies and proteome response”

=>Done

Line 24: What do the Authors mean by “absence of genes” in inbred lines? Which genes? Please clarify.

=>Done

Line 24: presence/absence of polymorphism

=>Done (sentence reformulated). Because of the limit of 200 words, few other changes were done in the abstract.

Line 73: Three traits are reported to be measured the day before the beginning of the treatment, 35 DAG in Table S1. However, in the Method’s section, the beginning of the treatment is reported at 38 DAG (line 475). Please explain or fix this inconsistency.

 =>Fixed. Sorry about that, there was a confusion between the day of water saturation and the one of beginning of stress.

Line 160: the category transcription is reported in Table S8 (and S6 and Figure 3) as RNA, please uniform.

=>Done

Line 162: I would say 2,3,7 and 11, from Table S8.

=>Yes you are right. We were taking in account several categories generally considered as involved in response to stress. For simplicity we deleted this sentence.

Line 217:

Table S1: I would suggest moving this Table to the main text, as it also reports all the abbreviations used in Fig.2. The number of eco-physiological traits here reported is 22 (column D), while 21 are reported in the title and in the main text (line 71). This likely because FTSW was just used as the measure of the watering state, but the information must be consistent.

=>Done. The table is now in the main text (Table 1). The title was changed to  “List of environmental and ecophysiologocal variables”

Heading of column E, ‘abbreviations’ is missing a ‘b’.

=>Done

I think the correct English abbreviation for number is No. or No, not Nb as in French (LNb_start). Same in Table S3, column H and I and elsewhere.

=>Thank you. Done in tables and figures.

Table S3: I think it would be more easy to follow the text if the three traits measured before the start of the treatment are reported separated from the other traits, i.e. in the first three row, or highlighted in some way (boldface or italics).

=>Done

Table S8: the total number of proteins in each Functional category could be indicated in a last column to simplify reading of the corresponding test.

=>Done.

 The last table line has no heading

=>Done

Names of Tables S8 and S9 are exchanged in the excel file

=>Done

Reviewer 3 Report (New Reviewer)

Article Sunflower hybrids and inbred lines adopt different proteomic and physiological strategies to cope with water deficit by authors

Harold Duruflé, Thierry Balliau, Nicolas Blanchet, Adeline Chaubet, Alexandra Duhnen, Nicolas Pouilly, Melisande Blein-Nicolas, Brigitte Mangin, Pierre Maury, Nicolas B Langlade, Michel Zivy describe a comparative analysis of the manifestations of water deficiency in the habitus of individual leaf blades and their response to lack of soil moisture. The article is of interest for this branch of research.

This manuscript is not designed according to the rules: 1) References and bibliography are not numbered; 2) there is no conclusion; 3) the materials and methods section does not provide detailed characteristics of plant cultivation, there is no diagram and photographs of plants that allow you to quickly understand how the experiment was carried out, how the plants looked, whether it corresponds to the state described by the authors, the conditions of air humidity, temperature, light, soil or substrate are not are given. References to other articles are not sufficient, as they do not allow the experiment to be reproduced in another laboratory.

More significant methodological errors: The title mentions proteomic strategies. It's hard for me to understand what the authors had in mind.

Sensitivity does not necessarily coincide with tolerance, and this is understandable. It is necessary to prove how plants behave when access to moisture is resumed based on the principle of reversibility.

Determination of parameters only in leaves cannot fully characterize the reaction of plants.

I think the authors should change the name, clearly formulate hypotheses, add photos, a diagram, expand the description of the experiment, taking into account the approach they proposed. Formulate a conclusion. Prepare the manuscript according to the rules.

Author Response

Article Sunflower hybrids and inbred lines adopt different proteomic and physiological strategies to cope with water deficit by authors

Harold Duruflé, Thierry Balliau, Nicolas Blanchet, Adeline Chaubet, Alexandra Duhnen, Nicolas Pouilly, Melisande Blein-Nicolas, Brigitte Mangin, Pierre Maury, Nicolas B Langlade, Michel Zivy describe a comparative analysis of the manifestations of water deficiency in the habitus of individual leaf blades and their response to lack of soil moisture. The article is of interest for this branch of research.

This manuscript is not designed according to the rules:

1) References and bibliography are not numbered;

=> Sorry about that. References and bibliography are now presented according to Biomolecules instructions to authors.

2) there is no conclusion
=> Conclusion added. This section is not mandatory according to the instructions to authors.

3) the materials and methods section does not provide detailed characteristics of plant cultivation, there is no diagram and photographs of plants that allow you to quickly understand how the experiment was carried out, how the plants looked, whether it corresponds to the state described by the authors, the conditions of air humidity, temperature, light, soil or substrate are not are given. References to other articles are not sufficient, as they do not allow the experiment to be reproduced in another laboratory.

=> All the details are extensively described in the data paper Blanchet et al (2018) and its supplemental data. The present manuscript is based on the experiment described it this data paper, as mentioned in the abstract (“exploiting publicly available datasets”). Dataset references are also given. To make things even clearer, the first sentence of the Material and Methods section is now :”Full details of growth conditions and measurement of eco-physiological traits are given in the data paper describing this experiment [51].”

In the same way, the first sentence of the desctiption of proteomics methods is now: “Full methods for sample preparation, mass spectrometry, protein identification and quantification are described in Balliau et al. [59] and in the data paper describing the data used in the present paper [60].”

As requested, we added a photograph in the supplemental data (Fig. S7), to help the reader to understand how the experiment was carried out.

More significant methodological errors: The title mentions proteomic strategies. It's hard for me to understand what the authors had in mind.

=> The title was changed to “Sunflower hybrids and inbred lines adopt different physiological strategies and proteome responses to cope with water deficit”

Sensitivity does not necessarily coincide with tolerance, and this is understandable. It is necessary to prove how plants behave when access to moisture is resumed based on the principle of reversibility.

=> We agree with reviewer 3 in that studying recovery would be very interesting. However recovery is a particular scenario. In the present paper, we focused on a simple and very common scenario.

Determination of parameters only in leaves cannot fully characterize the reaction of plants.

=> Measured ecophysological traits were not limited to leaves (see Table 1). However we agree with reviewer 3 in that studying the response of other organs would be very interesting, and we added the following sentence; “ However, further shared molecular mechanisms between plant species might be observed when studying non-photosynthetic organs involved in drought response in key organs such as roots and stems.”

I think the authors should change the name, clearly formulate hypotheses, add photos, a diagram, expand the description of the experiment, taking into account the approach they proposed. Formulate a conclusion. Prepare the manuscript according to the rules.

=> We changed the title, clarified the question about the description of the experiment, added a photo, and prepared the manuscript according to the instructions to authors.

Round 2

Reviewer 3 Report (New Reviewer)

Manuscript Article Sunflower hybrids and inbred lines adopt different proteomic and physiological strategies to cope with water deficit by

Harold Durufle et al. describe a comparative analysis of the manifestations of water deficiency in the habitus of individual leaf blades and their response to lack of soil moisture.

The article is of particular interest for this branch of research.

The shortcomings of the manuscript were eliminated.

This manuscript is a resubmission of an earlier submission. The following is a list of the peer review reports and author responses from that submission.

Round 1

Reviewer 1 Report

This work reports high-throughput data on eco-physiological traits and leaf proteomes from eight inbred lines of sunflower (Helianthus annuus) and their sixteen hybrids subjected to drought stress. The results are deeply analyzed using different statistical methods and detailed discussed, showing that hybrids respond more strongly to water deficit at the molecular and eco-physiological levels than their parental inbred lines. Based on these data, a reasonable molecular hypothesis which is proposed to explain such findings. The experimental design is appropriate to test the hypothesis, the methods are described in detail, the results are presented in a well-structured manner, and conclusions are consistent with the experimental evidence. In summary, this work is scientifically sound and brings new important data to this field of research. Some minor comments are shown below that should be addressed.

 Minor comments

1. Line 44. Please write sunflower scientific name in italics.

2. Lines 75-77. Carefully verify this part and amend it as needed: “While leaf area depended on the size of the leaf before the treatment, leaf expansion rate measured expansion over the last 24h and showed a significant reduction under WD in both mature and young leaves”.

3. Differentially abundant proteins (DAPs) identified by LC-MS/MS were distributed into 12 clusters. It would be helpful for readers not very familiar with proteome analysis if a short description could be provided on the criteria used to generate these clusters.

Reviewer 2 Report

Mechanism of drought tolerance in crops, in general and sunflower in particular, and the impact of hybridity status (hybrids vs inbred lines) on stress responses has been an interesting area of research. There are series of papers that develop the theme of water-deficit stress responses of sunflower encompassing recent developments at the physiological and molecular levels through agro-ecological transition and changes in protein expression [Angeles et al. (2008) Open Proteome J 1: 59-71; Fulda et al. (2011) Plant Biol 13: 632-642; Ghaffari et al. (2017) Crop Pasture Sci 68: 457-465]. The results presented, in this manuscript, reflects variations in genotypic levels and change their proteins expression that sunflower plants apply in adapting to the drought stress environment. However, there are few serious issues about this submission, which I have pointed out below:

Major Comments:

1.            The major drawback of this manuscript is that the same group recently published an article describing leaf-proteomic data set of 24 sunflower genotypes including both inbred lines and their hybrids [Balliau et al. (2021) OCL 28, 12, doi: 10.1051/ocl/2020074]. The authors also investigated the eco-physiological traits using the high-throughput phenotyping platform Phenotoul-Heliaphen as was performed in their previous study.

2.            The experimental design and plant materials in the previous work [Balliau et al. (2021) OCL 28, 12, doi: 10.1051/ocl/2020074] and the present study are the same. The authors previously reported the identification of 3062 proteins and the quantification of 1211 leaf proteins of the 24 genotypes grown under the identical watering conditions. Here they added the drought-responsive differentially accumulated. In addition, the authors included the genotyping data in this submission.   

3.            Most data describing the eco-physiological responses to water-deficit stress for these 24 genotypes are also available in the public domain [Balnchet et al. (2018) Data in Brief 21: 1296-1301].

4.            Although I didn’t use the plagiarism checker to calculate the percentage, but I’m sure that this is going to be significantly very high, which is a matter great concern.

Reviewer 3 Report

This manuscript contains new information concerning the relationship between heterosis and drought response in sunflower. Eight parental inbred lines and 16 hybrids were analysed for phenotypic traits and leaf proteome changes. Data are subjected to rigorous analysis and interesting conclusions are drown about different stress adaptive strategies in hybrids and inbred lines. The paper is well written. I have only a few remarks

The quality of figures could be improved.

Abbreviations in the text should be checked and properly introduced at first mention. Example: Line 85 – FTSW is an abbreviation, which is explained only at line 476 as fraction of transpirable soil water (FTSW). A list of abbreviations for physiological parameters is given in the supplementary but it is not enough.

Some references were not found in the text:

Balliau T, Duruflé H, Blanchet N, Blein-Nicolas M, Langlade NB, Zivy M. (2021). Proteomic data from leaves of  twenty-four sunflower genotypes underwater deficit. OCL 28, 12, doi: 10.1051/ocl/2020074.

Burstin J, Charcosset A. (1997). Relationship between phenotypic and marker distances: Theoretical and experimental investigations. Heredity 79: 477–483.

Chan KX, Wirtz M, Phua SY, Estavillo GM, Pogson BJ. (2013). Balancing metabolites in drought: The sulfur  assimilation conundrum. Trends in Plant Science 18: 18–29.

Edqvist J, Blomqvist K, Nieuwland J, Salminen TA. (2018). Plant lipid transfer proteins: Are we finally closing in on  the roles of these enigmatic proteins? Journal of Lipid Research 59: 1374–1382.

González I, Lê Cao KA, Davis M, Déjean S. (2013). Insightful graphical outputs to explore relationships between two ‘omics’ data sets. BioData Mining 5: 19.

Kleczkowski LA, Geisler M, Ciereszko I, Johansson H. (2004). UDP-Glucose Pyrophosphorylase. An Old Protein with New Tricks. Plant Physiology 134: 912–918.

Casadebaig P, et al. (2013). A biomarker based on gene expression indicates plant water status in controlled and natural  environments. Plant, Cell & Environment 36: 2175–2189.

Hopp HE, et al. (2017). Integration of transcriptomic and metabolic data reveals hub transcription factors involved in  drought stress response in sunflower (Helianthus annuus L.). Plant Molecular Biology 94: 549–564.

Ouvrard O, Cellier F, Ferrare K, Tousch D, Lamaze T, Dupuis J-M, Casse-Delbart F. (1996). Identification and expression of water stress- and abscisic acid-regulated genes in a drought-tolerant sunflower genotype. Plant Molecular  Biology 31: 819–829.

Rae GM, Uversky VN, David K, Wood M. (2014). DRM1 and DRM2 expression regulation: potential role of splice  variants in response to stress and environmental factors in Arabidopsis. Molecular Genetics and Genomics 289: 317–332.

Wang Y, Yang L, Chen X, Ye T, Zhong B, Liu R, Wu Y, Chan Z. (2016). Major latex protein-like protein 43 ( MLP43 )  functions as a positive regulator during abscisic acid responses and confers drought tolerance in Arabidopsis thaliana.  Journal of Experimental Botany 67: 421–434.

Xu X, Yang Y, Liu C, Sun Y, Zhang T, Hou M, Huang S, Yuan H. (2019). The evolutionary history of the sucrose  synthase gene family in higher plants. BMC Plant Biology 19: 1–14. 

Xu Y, Zheng X, Song Y, Zhu L, Yu Z, Gan L, Zhou S, Liu H, Wen F, Zhu C. (2018). NtLTP4, a lipid transfer protein that  enhances salt and drought stresses tolerance in Nicotiana tabacum. Scientific Reports 8: 8873.